# Assessing the relevance and uptake of core outcome sets (an agreed minimum collection of outcomes to measure in research studies) in Cochrane systematic reviews: a review

Paula R Williamson,[1] Ricardo de Ávila Oliveira,[2] Mike Clarke [ID],[3] Sarah L Gorst [ID],[1] Karen Hughes,[1] Jamie J Kirkham [ID],[4] Tianjing Li,[5] Ian J Saldanha,[6] Jochen Schmitt[7]

For numbered affiliations see end of article.

**Correspondence to**
Dr Paula R Williamson;
prw@liv.ac.uk

## ABSTRACT

**Objectives** A core outcome set (COS) is an agreed standardised minimum collection of outcomes that should be measured and reported in research in a specific area of health. Cochrane systematic reviews ('reviews') are rigorous reviews on health-related topics conducted under the auspices of Cochrane. This study examines the use of existing COS to inform the choice of outcomes in Cochrane systematic reviews ('reviews') and investigates the views of the coordinating editors of Cochrane Review Groups (CRGs) on this topic.

**Methods** A cohort of 100 recently published or updated Cochrane reviews were assessed for reference to a COS being used to inform the choice of outcomes for the review. Existing COS, published 2 or more years before the review publication, were then identified to assess how often a reviewer could have used a relevant COS if it was available. We asked 52 CRG coordinating editors about their involvement in COS development, how outcomes are selected for reviews in their CRG and their views of the advantages and challenges surrounding the standardisation of outcomes within their CRG.

**Results** In the cohort of reviews from 2019, 40% (40/100) of reviewers noted problems due to outcome inconsistency across the included studies. In 7% (7/100) of reviews, a COS was referenced in relation to the choice of outcomes for the review. Relevant existing COS could be considered for a review update in 35% of the others (33/93). Most editors who responded (31/36, 86%) thought that COS should definitely or possibly be used to inform the choice of outcomes in a review.

**Conclusions** Systematic reviewers are continuing to note outcome heterogeneity but are starting to use COS to inform their reviews. There is potential for greater uptake of COS in Cochrane reviews.

## BACKGROUND

Systematic reviews ('reviews') of the effects of healthcare interventions summarise the available evidence and are used to inform decision-making. Inconsistencies in the outcomes that

### Strengths and limitations of this study

► A sample of 100 Cochrane systematic reviews that were published for the first time or updated in 2019 was assessed.
► Two investigators independently assessed the relevance of a core outcome set (COS) to each systematic review.
► An assumption was made that if a COS had been published 2 or more years before the publication of the systematic review, it would have been available to the reviewers to adopt.
► Coordinating editors for all Cochrane Review Groups were asked for their views on the relevance of COS to systematic reviews.
► The research was undertaken before the release of V.6 of the Cochrane Handbook for Systematic Reviews of Interventions, which recommends that Cochrane reviewers should consider using a COS.

are measured and reported across studies included in reviews can mean that it is not possible for all studies to be included in a meta-analysis; however, a form of research waste. In addition, the selective reporting of outcomes by some included studies based on the direction or magnitude of the results, a phenomenon known as outcome reporting bias, can affect the robustness of the conclusions of a review.[1 2]

Furthermore, there is an increasing awareness that differences exist between the outcomes measured in clinical trials and outcomes that patients consider important. For example, only 50% of the outcomes that patients said matter to them were captured in the clinical trials reviewed by the Canadian Agency for Drugs and Technologies in Health

Common Drug Review.[3] Such problems also contribute to research waste.[4]

One solution to these problems with the choice of outcomes is for clinical trialists in a particular topic area to measure and report, at a minimum, a core outcome set (COS), which would then be used in reviews addressing that same topic area. A COS is an agreed standardised collection of outcomes that should be measured and reported in a specific area of health. It does not preclude the inclusion of additional outcomes, but represents the minimum for all research studies in the topic area. The scope of a COS refers to the specific area of health or healthcare of interest to which the COS is to be applied, defined in terms of the health condition, population and healthcare interventions covered. Thus, within a particular topic area there may be multiple COSs, depending on whether a COS is developed for different populations, for example adults versus children, or localised versus advanced disease. A COS may be developed to apply to any intervention for that condition, or it may be for specific types, for example surgery.

Minimum standards have been established for developing COS.[5] According to these standards, as a minimum, the stakeholders who should be involved in developing a COS should be those with lived experience of the condition, health professionals caring for those with lived experience and those undertaking research in the condition.[5]

The benefits of widespread adoption of a COS are to increase outcome consistency across trials, resulting in a major reduction in selective reporting and maximising the potential for a trial to contribute to meta-analyses of these key outcomes.[6] Of crucial importance is that, given the expectations of relevant stakeholder involvement, the use of a COS should mean trialists are much more likely to measure appropriate outcomes that are relevant to patients.

An additional issue in the choice of outcomes for a systematic review relates to clinical trialists and systematic reviewers sometimes being interested in different types of outcomes.[7 8] Clarke and Williamson have argued for greater involvement of systematic reviewers in the development and implementation of COS,[1] suggesting this might help with planning of reviews and ensuring that reviews identify data for outcomes of interest.

Cochrane reviews are rigorous systematic reviews conducted under the auspices of Cochrane, a global and independent volunteer organisation with the primary goal of conducting evidence syntheses in health. Cochrane is organised operationally into Cochrane Review Groups (CRGs), comprised of an editorial team, information specialists, and authors responsible for reviews within distinct clinical and public health areas. For specific reviews, decisions need to be made regarding outcomes that will be included in the review as well as the subset of key outcomes that will be prioritised for inclusion in the summary of findings (SoF) tables. SoF tables are intended to provide consumers of the review with the key findings of the review.

Each CRG is led by a coordinating editor who oversees the production and maintenance of Cochrane reviews. In a 2012 survey of CRG coordinating editors (hereafter 'editors'), the majority (33/45, 73%) thought that COS should routinely be used in the SoF tables.[9] At that time, a third of the editors (14/45, 31%) reported personal involvement in the development of a COS.

The awareness and development of COS have substantially increased over recent years, with more than 100 new COS published between 2012 and 2019 and at least 200 more in development.[10–16] A series of guidelines have been published to improve the design and reporting of COS development.[17–20] The relevance of COS to defining review questions and planning the review, as well as the inclusion of patient-centred outcomes within COS, is acknowledged in the 2019 version of the Cochrane Handbook for Systematic Reviews of Interventions, which states, 'where available, established sets of core outcomes should be used'.[21] One notable link between COS development and Cochrane is the work of the Cochrane Skin Group on the Cochrane Skin Core Outcome Set Initiative (CS-COUSIN).[22 23] The mission of this group is to standardise outcomes in dermatology clinical trials in order to make trial evidence more comparable and, thereby, to strengthen the quality, interpretability and ability of systematic reviews to facilitate evidence-based decision-making in dermatology.

In this paper, we examine current practices of Cochrane systematic reviewers in relation to the use of COS in choosing outcomes, and the current views of CRG editors regarding the adoption of COS in their CRGs.

## METHODS
### Assessment of Cochrane reviews in relation to COS
Although a formal protocol was not developed for this study, the following approach was agreed among the research team in advance. We examined the first 100 new or updated Cochrane intervention reviews published in 2019 (dated 1 January to 8 March). We chose to restrict to reviews published in 2019 in order to determine contemporary practices regarding outcome choice in Cochrane reviews. The sample size was a pragmatic choice, to give a reasonable number of reviews that we could assess in a reasonable period of time.

We examined whether, in the completed version of the review, the review authors: (1) mentioned using a COS to choose outcomes for the review, even if all the outcomes in the COS may not have been used; (2) identified any problems with outcome inconsistency across the studies included in the review and (3) noted the need for development of a COS. We extracted information that may have been reported in any section of the completed review. One investigator (among IJS, JJK, KH, PRW, RdAO and TL) extracted information from each review and one investigator (between MC and IJS) verified the extracted information.

For each review, we searched the Core Outcome Measures in Effectiveness Trials (COMET) Database (a regularly updated online repository of COS studies) to establish whether a COS with a relevant scope (even if not an exact match) had been published by 31 December 2016, such that it might have been available to inform outcome choice at the planning stage for reviews that were published more than 2 years later. For reviews where no such COS was found, the COMET Database was also searched for COS published since 1 January 2017, to determine whether a relevant COS may be available for consideration when the Cochrane review is updated. Two investigators (SLG and PRW) assessed the potential relevance of identified COS independently and then compared their assessments, following an approach used previously (data available on request).[24]

### Coordinating editors of CRGs

Consistent with the approach used in our 2012 survey,[7] we emailed the editors of all 52 CRGs (as of 2019) requesting them to provide information on their involvement with COS development and their CRG policy with regards to outcome selection for reviews and prioritisation of outcomes for associated SoF tables. They were presented with a breakdown of the total number of completed and ongoing COS for research, with references, that were within the scope of their CRG (figure 1 summarises the number of COS by topic area).

This list of COS was compiled from the annual update of a review of COS studies conducted by the COMET initiative and was last updated in 2018.

Editors were also asked about their opinions on the standardisation of outcomes within their CRG, including what they thought were up to three main advantages and three main challenges of standardising outcomes across all reviews in a particular condition covered by their CRG. They were provided with the responses from the 2012 survey for their CRG and were asked to indicate whether those earlier responses remained the same or to identify any changes of opinion.

Expanding on our 2012 survey, we requested them to provide details of any examples where a COS had been considered to guide the choice of outcomes to include in a Cochrane review or a SoF table. Second, we described the aims of the CS-COUSIN initiative and asked editors for their views on the relevance and feasibility of establishing links between their CRG and an equivalent COS group.

Responses were collected between 26 February and 30 April 2019. Non-responders were contacted every 3 weeks during this period with reminders to participate.

### Analysis

The data from the assessment of Cochrane reviews were analysed descriptively. The lists of advantages and challenges of COS suggested by editors was independently reviewed by two authors (JJK and PRW) and coded using the categories identified in the 2012 survey, allowing new categories to be added. Discrepancies in categorisation were resolved through discussion.

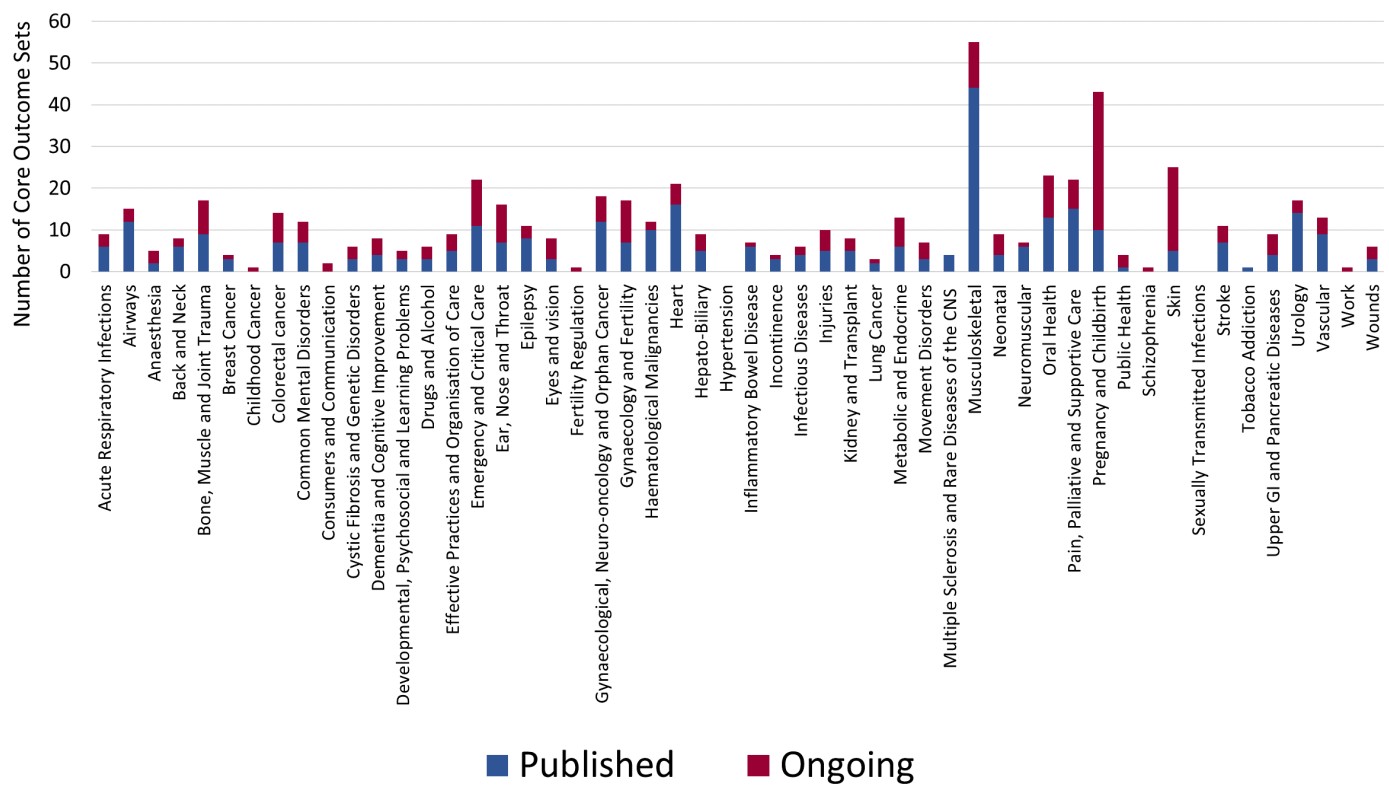

**Figure 1** Published and ongoing core outcome sets according to Cochrane Review Group.

## Patient and public involvement

Patients or the public were not involved in the design, conduct or reporting of this research.

## RESULTS

### Assessment of Cochrane reviews in relation to use of COS to inform outcome choice

The reviews assessed addressed a range of health conditions (table 1). Forty-one of the 52 current CRGs (79%) were represented. Forty reviews (40%) identified problems of inconsistency in outcomes in included studies; in five of these, the authors also explicitly noted the need for development of a COS.

Seven of 100 (7%) reviews mentioned (or cited) a COS in relation to choosing outcomes for the review. A further three reviews, while not mentioning a COS in relation to choosing outcomes for the review and not specifically recommending the use of a COS, made observations about existing COS. One review noted that several patient-important outcomes identified in an existing COS were absent from most of the review's included studies. One review acknowledged an existing COS, noting that the trials in the review continued to report inconsistent outcomes, and noted that further work was needed on how the outcome domains in the COS should be measured to facilitate future reviews. The third review recommended that researchers should adopt the definition of a particular outcome domain of interest to the review as provided in an existing COS.

For the 90 reviews not referring to a COS, a relevant COS was found to have been published in 2016 or earlier in 24 reviews. Thus, of 93 reviews not referencing a COS in relation to choosing outcomes for the review, in 27 (29%) cases, this could have been possible.

In summary, of the 100 reviews, a COS from 2016 or earlier was available for 34 reviews (34%). Of those 34 reviews, a COS was used in 7 reviews (21%). In a further six reviews, a relevant COS was published in 2017 or later. Thus, a relevant COS could be considered during the updating of at least 35% (33/93) of the reviews that have not already used a COS.

### Editors' views

Thirty-eight (73%) of the 52 CRG editors responded to our emails, although some did not provide a response to all questions. Two-thirds of the editors (23/35, 66%) had been involved in the development of a COS, 8 editors prior to the 2012 survey and 15 since then.

As of 23 April 2019, the COMET Database contained 323 published and 242 ongoing COS across all CRGs, ranging from 0 (hypertension and sexually transmitted infections) to 54 (musculoskeletal) (figure 1). The total number of published and ongoing COS was slightly higher for CRGs whose editors responded in 2019 (median 9, IQR 5–16.5) compared with not (median 7, IQR 5–9).

Of 22 editors who responded in both 2012 and 2019, 64% noted in 2012 that COS should be used in SoF; this percentage rose to 91% in 2019.

**Table 1** Health conditions addressed in the 100 Cochrane reviews assessed in relation to use of COS

| ICD-11 code | Disease area | N (%) |
|---|---|---|
| 12 | Respiratory system | 11 (11) |
| 16 | Genitourinary system | 11 (11) |
| 2 | Neoplasms | 9 (9) |
| 11 | Circulatory system | 8 (8) |
| 1 | Infectious or parasitic diseases | 5 (5) |
| 8 | Nervous system | 5 (5) |
| 13 | Digestive system | 4 (4) |
| 15 | Musculoskeletal system | 4 (4) |
| 9 | Visual system | 4 (4) |
| 14 | Skin | 3 (3) |
| 24 | Factors influencing health status or contact with health services | 3 (3) |
| 4 | Immune system | 2 (2) |
| 5 | Endocrine, nutritional or metabolic diseases | 2 (2) |
| 6 | Mental, behavioural or neurodevelopmental | 2 (2) |
| 7 | Sleep–wake disorders | 2 (2) |
| 17 | Conditions related to sexual health | 2 (2) |
| 3 | Blood or blood-forming organs | 1 (1) |
| 10 | Ear or mastoid process | 1 (1) |
| 19 | Certain conditions originating in the perinatal period | 1 (1) |
| 21 | Symptoms, signs or clinical findings, not elsewhere classified | 1 (1) |
| 22 | Injury, poisoning or certain other consequences of external causes | 1 (1) |
| 23 | External causes of morbidity or mortality | 1 (1) |
| No specific ICD-11 code | Miscellaneous | 17 (17) |

ICD-11, International Classification of Diseases 11th Revision.

### CRG outcome policies

Figures 2 and 3 show the policies of CRGs for outcome selection for reviews (figure 2) and for SoF tables (figure 3); both show a reduction in a policy of authors' discretion since 2012, to 49% in 2019. Currently, 40% (14/35) of CRGs have a centralised policy for both review and SoF table outcome choice. Four editors (11%) indicated that outcome choice was now a negotiation between authors and the editorial team, with input from peer reviewers.

In 2019, half of the editors (18/36; 50%) thought that COS should definitely be considered in the process of choosing outcomes for an SoF, while a further 13 (36%) editors thought that COS could possibly be used in some

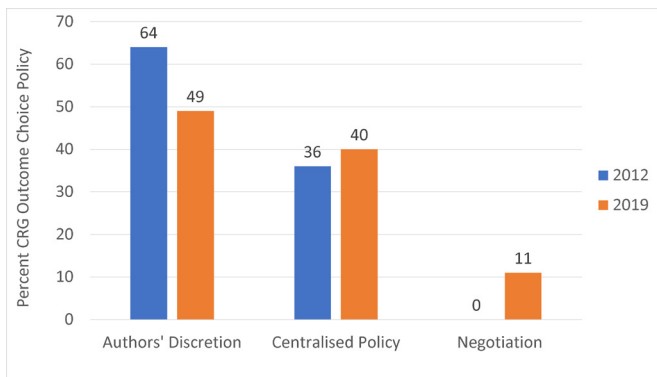

**Figure 2** CRG outcome choice policies. Denominators are (n=45) and (n=35) for 2012 and 2019, respectively. CRG, Cochrane Review Group.

circumstances if relevant to the specific review and if the COS was developed using a 'quality' consensus approach. Two (6%) editors thought that COS should not be used in the SoF tables because they can lead to the inclusion of unnecessary outcomes. One editor did not think that the inclusion of COS in SoF tables was relevant, because they did not think COS were used in their topic area. Two editors were unsure because they were aware that many COS contain more than the maximum of seven outcomes that Cochrane allows for SoF tables.

### Advantages and challenges of standardising outcomes in Cochrane reviews

For both the advantages and challenges associated with standardising outcomes across reviews (table 2), the three most commonly listed were the same as in the 2012 survey. Just under half of the editors consider an advantage of using a COS to be that the outcomes are likely to be more appropriate.

One notable difference from 2012 was a shift from the most frequently listed challenge being COS development (55% in 2012, falling to third place on 22% in 2019) to deciding when a COS should be applied (in relation to the scope of the review and the scope of the COS) in 2019 (78%). One new challenge identified in 2019 by three editors (8%) was the perception that the use of a

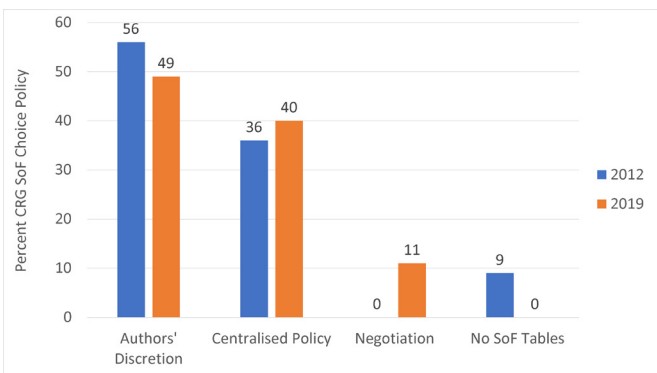

**Figure 3** CRG SoF table choice policies. Denominators are (n=45) and (n=35) for 2012 and 2019, respectively. CRG, Cochrane Review Group; SoF, summary of findings.

**Table 2** Views of Cochrane editors on standardising outcomes across all reviews for a particular condition

| Advantages | 2012 n/40 (%*) | 2019 n/32 (%*) |
| --- | --- | --- |
| Advantage for a systematic review/meta-analysis | 39 (98) | 27 (84) |
| Improves interpretation/guidance | 19 (48) | 7 (22) |
| Outcome likely to be more appropriate | 16 (40) | 10 (45) |
| Advantage for the design of a new study | 13 (33) | 5 (16) |
| Improves something about the outcome itself (eg, simplifies the reporting) | 6 (15) | 0 (0) |
| Reduces outcome reporting bias | 6 (15) | 1 (3) |
| Reduces resource requirement (eg, time to review) | 1 (3) | 1 (3) |
| **Challenges** | **2012 n/42 (%*)** | **2019 n/36 (%*)** |
| Development of a COS | 23 (55) | 8 (22) |
| Something about scope | 21 (50) | 28 (78) |
| How to persuade authors/trialists/industry to implement | 20 (48) | 14 (39) |
| 'How' to measure once the 'what' has been decided | 11 (26) | 3 (8) |
| Important outcomes not currently being measured | 2 (5) | 1 (3) |
| Resource to develop | 2 (5) | 3 (8) |
| Updating process | 2 (5) | 2 (6) |
| Conflict of interest | 1 (1) | 0 (0) |
| Limits authors | 0 (2) | 3 (8) |

*Percentages represent the number of Cochrane Review Group coordinating editors who mentioned each advantage/challenge. COS, core outcome set.

COS may limit review authors due to a lack of flexibility in outcome choice.

### Links between CRG, COS developers and guidance

Eight editors identified cases where the outcome choice in at least one review in their group had been informed by a COS. The majority (22/26, 85%) of editors thought that links between their CRG and COS developers in their field would be a good idea in principle, while four could see no benefit. In addition to the Cochrane Skin Group, three CRGs had already established such links: the Cochrane Musculoskeletal Group with the Outcome Measures in Rheumatology initiative,[25] the Cochrane Kidney and Transplant Group with the Standardising Outcomes in Nephrology initiative[26] and the Cochrane Pregnancy and Childbirth Group with the Core Outcomes in Women's and Newborn Health initiative.[27]

The major perceived barriers to establishing links between CRGs and COS developers were related to

lack of resources, mainly lack of time (mentioned nine times) and funding to coordinate (mentioned five times). Conflicts of interest, the potential for poor communication between members and lack of knowledge on how to manage such links were also quoted as potential barriers to implementation.

## CONCLUSIONS

Our study has demonstrated that Cochrane systematic reviewers note widespread outcome inconsistency across studies. We found that almost all CRG editors think that COS should definitely or possibly be used in SoF tables, and that reviewers are beginning to use COS to inform their choice of outcomes.

We have documented an increased level of involvement of CRG editors in COS development studies since 2012. The COS-STAD (Core Outcome Set-Standards for Development) guidance identifies three stakeholder groups as the minimum for input into the development of a COS: patients or their representatives, healthcare professionals and those who will use the COS in research.[17] Consensus was not quite achieved that a fourth group, those who will use the research that should use the COS (eg, systematic reviewers, guideline developers, policy makers and regulatory agencies), should always be involved. COS-STAD participants may have wanted to avoid setting a minimum standard when there was limited experience of engaging with these particular stakeholder groups. If the involvement of Cochrane reviewers in COS development studies continues to increase, it would seem appropriate to review the standard in due course.

The majority of editors were in favour of cooperation between their CRG and COS developers in their field; however, the most frequently mentioned potential barriers, such as lack of time, resources and funding, will need to be addressed to achieve this. The affiliation of a COS development initiative to a CRG, as with CS-COUSIN in dermatology,[22 23] could offer easier access to information about outcomes previously measured in trials, and to Cochrane's international network of consumers, healthcare professionals and researchers in the field. Additionally, close interaction between COS development and CRGs should support COS implementation. For the dermatology outcome research community, the formal affiliation of COS development groups to Cochrane is considered an important advantage. This strategy has led to the affiliation of 15 COS groups to CS-COUSIN since its initiation in 2015.[22 23]

Our study has some limitations. No formal protocol was developed for this study; however, the research team had agreed the methodological approach prior to commencing the study.

Second, we achieved a 73% response rate from CRG coordinating editors. While this rate is acceptable, it is conceivable that non-response bias may have arisen as a result of a lack of interest or awareness about COS among editors not responding. A third limitation may have been our assumption that if a COS had been published 2 or more years before the publication of the systematic review, it would have been available to the reviewers to adopt. This may not have always been true. Previous studies have found that reviews generally,[28] and Cochrane reviews specifically,[29] take an average of less than 2 and 2.4 years to be completed, respectively. This suggests that our assumption regarding availability of the COS to the systematic reviewers may not be unreasonable.

Cochrane now recommends that reviewers should consider using COS where these exist.[18] Our study identifies some implications for practice when following this guidance. The COMET database of COS (available online at http://www.comet-initiative.org/) is free, searchable and covers a range of health fields. Using a given COS for a given systematic review involves, at the least, an assessment of the appropriateness of the topic, scope, stakeholder representation and currency of the COS. We recommend that, when a potentially appropriate COS exists for a given systematic review, the authors should either use it to inform their choice of outcomes or justify their reason for not using it. In addition, authors of Cochrane reviews, especially those that identify outcome inconsistency in included studies, should take the opportunity to move their respective fields forward by explicitly noting the need for COS, recommending COS development and participating in the COS development process.

Our study has some implications for research. The assessment presented here will act as a benchmark for subsequent evaluation of the implementation of the guidance in the Cochrane Handbook that is specific to using COS when choosing outcomes for Cochrane reviews. We note that 66% of the review topics, from 31 CRGs, did not have a relevant COS. Future research should include identification of priority areas for COS development.

The use of COS for a systematic review in turn may influence clinical trialists to use that same COS to ensure that their trials can contribute fully to subsequent reviews and meta-analyses. Given the continued problem of outcome inconsistency in studies documented in Cochrane reviews, and the fact that four CRGs have already established links to COS development groups, this topic could be a potential area for strategic development of Cochrane. Such greater uptake of COS across the healthcare research ecosystem could help improve research and thereby improves healthcare and health.

**Author affiliations**

[1]MRC North West Hub for Trials Methodology Research, University of Liverpool and member of Liverpool Health Partners, Liverpool, UK

[2]DECIR Faculdade de Medicina, Universidade Federal de Uberlândia, Uberlândia, Brazil

[3]Northern Ireland Methodology Hub, Centre for Public Health, Queen's University Belfast, Belfast, UK

[4]Centre for Biostatistics, Manchester Academic Health Science Centre, University of Manchester, Manchester, UK

[5]Department of Ophthalmology, University of Colorado Denver, Denver, Colorado, USA

[6]Center for Evidence Synthesis in Health, Brown University School of Public Health, Providence, Rhode Island, USA

[7]Center for Evidence-based Healthcare, Medizinische Fakultät, Technische Universität Dresden, Dresden, Germany

**Acknowledgements** We thank all the Cochrane Review Group coordinating editors who responded to our questions. We are grateful to the reviewers for their comments, which have helped us to improve our manuscript.

**Contributors** PRW conceived the idea for the manuscript. JJK, MC, JS, and PRW developed the questions for the Cochrane Review Group (CRG) coordinating editors. The list of the total number of completed and ongoing core outcome set (COS) with references for each CRG was produced by SLG. The requests to the editors in 2019 were administered by JJK. Their responses were analysed by JJK in collaboration with PRW. The review of Cochrane reviews was designed by IJS and PRW. The data were extracted by RdAO, KH, JJK, TL, IJS and PRW, confirmed by MC, and analysed by IJS. The assessment of relevance of published COS was undertaken by SLG and PRW. PRW, JJK and IJS prepared the initial manuscript. All authors commented on and approved the final manuscript before submission.

**Funding** This work was supported by an Medical Research Council (MRC) Hub for Trials Methodology Research Network studentship (grant reference MR/L004933/1-Q30); a National Institute for Health Research (NIHR) Senior Investigator Award (NF-SI_0513-10025); and the MRC Trials Methodology Research Partnership (grant reference MR/S014357/1). The views expressed in this article are those of the authors and not necessarily those of the NIHR, or the Department of Health and Social Care.

**Competing interests** PRW and MC are members of the COMET (Core Outcome Measures in Effectiveness Trials) Management Group. JJK and JS are members of the CS-COUSIN (Cochrane Skin—Core Outcome Set Initiative) methods group. The remaining authors declare no competing interests.

**Patient and public involvement** Patients and/or the public were not involved in the design, or conduct, or reporting, or dissemination plans of this research.

**Patient consent for publication** Not required.

**Provenance and peer review** Not commissioned; externally peer reviewed.

**Data availability statement** All data are available from the corresponding author on reasonable request.

**Open access** This is an open access article distributed in accordance with the Creative Commons Attribution 4.0 Unported (CC BY 4.0) license, which permits others to copy, redistribute, remix, transform and build upon this work for any purpose, provided the original work is properly cited, a link to the licence is given, and indication of whether changes were made. See: https://creativecommons.org/licenses/by/4.0/.

**ORCID iDs**
Mike Clarke http://orcid.org/0000-0002-2926-7257
Sarah L Gorst http://orcid.org/0000-0002-7818-9646
Jamie J Kirkham http://orcid.org/0000-0003-2579-9325

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
