## [Reviewer comments · BMJ Open]

ARTICLE DETAILS

TITLE (PROVISIONAL)	Assessing the relevance and uptake of core outcome sets (an agreed minimum collection of outcomes to measure in research studies) in Cochrane systematic reviews: a review
AUTHORS	Williamson, Paula; de Ávila Oliveira, Ricardo; Clarke, Mike; Gorst, Sarah L.; Hughes, Karen; Kirkham, Jamie J.; Li, Tianjing; Saldanha, Ian; Schmitt, Jochen

VERSION 1 – REVIEW

REVIEWER	Andy Oxman Norwegian Institute of Public Health I have in the past (over 5 years ago) collaborated with one of the authors.
REVIEW RETURNED	22-Jan-2020

GENERAL COMMENTS	This is a well written report. My only suggestion is that you might want to consider expanding on limitations of the study, implications for practice, and implications for research.
---

REVIEWER	Ray Moynihan Institute for Evidence-Based Healthcare, Bond University, Australia I am a supporter of Cochrane, and was contracted to present a podcast by Cochrane Australia.
REVIEW RETURNED	18-Feb-2020

GENERAL COMMENTS	General Thank you for the opportunity to review this paper. I should state upfront that I am a long time supporter of the Cochrane collaboration, and thus I am well disposed to research aimed at improving the quality of Cochrane reviews. However, my main concerns with the current version of the manuscript is that it lacks clarity and is more promotional than academic in its tone. The main concern is a lack of clarity. I give some small examples below in my specific comments – but in general, the manuscript reads like it is written as a report to a Cochrane executive about COS, rather than a study written up for a general medical audience. I think the manuscript needs to be written much more for a general readership, who may know very little about Cochrane Reviews,
---

and about COS. There needs to be much more of an attempt to explain all terms, and particularly to explain the meaning and significance and value of using COS.

The second main problem is the promotional tone – which comes through particularly strongly in the Conclusions section. This section reads more like a promotional advocacy for embracing COS than the Discussion section of an academic journal, discussing the findings of a study. Apologies if I missed it, but I could see no Limitations addressed in this section.

I think any revision would benefit greatly from being much clearer, and much less promotional.

Also, forgive me if I missed it, but I saw no mention of a protocol for the study – and it would be good to include a protocol as a supplement, and/or explain why there was no protocol. Similarly, I am unsure whether there is a relevant reporting checklist, but if so, that may be needed. (forgive me if I missed it)

Detailed comments:

Title:

I wonder whether a clear title could be found – it's not immediately clear what a "core outcome set" is.

Abstract

Page 3, Line 41: This opening sentence needs to be clearer – I have read it a few times and still don't quite understand it. I wonder too whether "core outcome set" needs to be briefly explained (as happens in paragraph two of the Background section) It's not immediately clear what it is.

Methods

Page 3- Lines 45-51. I think this section could be clearer. I don't understand how or why results were compared with existing COS. Its not clear what "CRG outcome selection policies" are. Introducing the notion of a "Summary of Findings" table also adds complexity and confusion. It feels like it is written with an assumption that the reader will understand all of this complexity, but I think it could be written more clearly, in a way that is comprehensible to people unfamiliar with these complexities.

Results:

Page 3- Lines 53- 57. I also find this section very hard to understand – with similar comments to my comments about Methods section of Abstract. For example I have no idea what the sentence means stating "In 7%..." I may well understand all of this clearly by the time I have read the full text – but I think the abstract needs to be much clearer.

Similar concerns about clarity and comprehension for "Strengths and Weaknesses"

Background

Page 5- Line 91. Is it possible to say why this affects the robustness of conclusions? In order to add clarity and underscore significance of this study.

	First paragraph: Page 5, Lines 88-96. There is too much going on in this paragraph that needs more explanation. The Background section needs to be clearer and be able to impart clearly the process of a review, how/why reviewers chose certain outcomes etc Methods Page 7 – Lines 141-143. This sentence could be clearer – eg what does ‘relevant scope’ mean? Also it is not clear from the text why the years 2016, and the time frame 2017 or later – were chosen. Page 7 – Lines 146- 147. It is not clear what “compared results” means. Page 7, Line 152 – for this phrase “outcome selection for the review and for its SoF tables” – do you mean for the reviews conducted within their CRG? Its not clear. Page 7 – Lines 153 -155 and Figure 1. Forgive me but I am uncertain why there are so many Core Outcome Sets within each condition category – and how the many COS interact with each other. I can presume why there are so many – but perhaps it might be explained more clearly in the text. Perhaps I have misunderstood the meaning/purpose of COS – and perhaps in the revision this can be explained more clearly and simply. Results Page 10- Lines 208-210. While I have no question about accuracy – as a reader it is a little confusing jumping from the “90” in the first sentence of this paragraph, to the “93” in the next sentence. Conclusions Whole tone of the Conclusion needs to change to become more like an academic paper and less like a promotional piece promoting the value of COS.
--	--

REVIEWER	Spyridon N. Papageorgiou University of Zurich, Switzerland
REVIEW RETURNED	06-Apr-2020

GENERAL COMMENTS	This is a simple and interesting paper on an interesting matter. I have written a few things that are very minor and am supportive of the authors’ work. Thanks for letting me see this.  • Short clarification on the sample size used: I take it that 100 was used arbitrarily or were there practical reason. Maybe add a couple of words to clarify this. • In “(i) mentioned using a COS to choose outcomes for the review”: are we talking about the protocol stage of the review or the write-up (including post hoc adaptations of the registered protocol). • Can you elaborate what is meant in (ii) with problems in outcome inconsistency? • Regarding the ‘data available on request’ on page 7: I generally am not a great fan of such statements. You might consider if this info can fit in an online appendix and provided a priori. • Maybe you can add a short sentence explaining how is each CRG editor involved in what phases of a review, so that readers
---

	not familiar with the structure of Cochrane can understand it better?  • I am skeptical about the use of 2016 or 2017 as chosen years for the need for a COS to have been published in order for the reviewer to be able to include them. Is it based on an average preparation duration for a review of 2 years (as you searched for reviews published in 2019)? And even if that is true, it might be that the reviewers might not have seen a published COS at the protocol phase, but they could have seen it at a later stage during analysis or write-up. Anyway, just to give this some consideration as an issue in the manuscript, if possible. • “Of 34 reviews where a COS was available (2016 or earlier), one was used in 7 (21%).”: I found the placement of this statement here somewhat confusing (then I understood that it adds up the 27 reviews from directly above to the 7 reviews at the top of the page...Maybe consider modifying it to make it more straightforward? • Ok, this is a standard term, but maybe explain the IQR in its first mention. • There was some vagueness as to where the Results section and the Discussion section ends/starts, but I don't know exactly the journal's rules. I personally did not find this a problem and thought the text flow was smooth from reporting of results to discussion of issues. I am also not that sure how much pertinent literature exists in this specific field (I take it not that much and the authors are probably already aware of it). • I believe the 1st figure with the COSs is difficult to be seen on paper without magnification option. • For the next 2 figures, maybe consider differentiating the two years with different columns or colour?
--	---

VERSION 1 – AUTHOR RESPONSE

Reviewer: 1

Reviewer Name: Andy Oxman

Institution and Country: Norwegian Institute of Public Health Please state any competing interests or state 'None declared': I have in the past (over 5 years ago) collaborated with one of the authors.

Please leave your comments for the authors below This is a well written report. My only suggestion is that you might want to consider expanding on limitations of the study, implications for practice, and implications for research.

- We have discussed the limitations of our study in more detail.
- We have expanded the existing paragraph in the Conclusions section on implications for practice, and added some implications for research.

Reviewer: 2

Reviewer Name: Ray Moynihan

Institution and Country: Institute for Evidence-Based Healthcare, Bond University, Australia Please state any competing interests or state 'None declared': I am a supporter of Cochrane, and was contracted to present a podcast by Cochrane Australia.

Please leave your comments for the authors below

General

Thank you for the opportunity to review this paper. I should state upfront that I am a long time supporter of the Cochrane collaboration, and thus I am well disposed to research aimed at improving the quality of Cochrane reviews.

However, my main concerns with the current version of the manuscript is that it lacks clarity and is more promotional than academic in its tone.

The main concern is a lack of clarity. I give some small examples below in my specific comments – but in general, the manuscript reads like it is written as a report to a Cochrane executive about COS, rather than a study written up for a general medical audience.

I think the manuscript needs to be written much more for a general readership, who may know very little about Cochrane Reviews, and about COS. There needs to be much more of an attempt to explain all terms, and particularly to explain the meaning and significance and value of using COS.

– We have added text to the Background to address these comments.

The second main problem is the promotional tone – which comes through particularly strongly in the Conclusions section. This section reads more like a promotional advocacy for embracing COS than the Discussion section of an academic journal, discussing the findings of a study. Apologies if I missed it, but I could see no Limitations addressed in this section.

– We have revised the text in the Conclusions section, and discussed the limitations of our study in more detail.

I think any revision would benefit greatly from being much clearer, and much less promotional.

Also, forgive me if I missed it, but I saw no mention of a protocol for the study – and it would be good to include a protocol as a supplement, and/or explain why there was no protocol. Similarly, I am unsure whether there is a relevant reporting checklist, but if so, that may be needed. (forgive me if I missed it)

– No formal protocol was developed for this study, although the methods were agreed by the research team in advance; this note has been added to the text. The methods used in the previous 2012 survey of Cochrane CRG Editors were followed, and this note has been added to the beginning of the Methods section. To our knowledge, there is no reporting guideline for the type of work presented in this paper.

Detailed comments:

Title:

I wonder whether a clear title could be found – it's not immediately clear what a "core outcome set" is.

– We have expanded the title to "Assessing the relevance and uptake of core outcome sets (an agreed minimum collection of outcomes to measure in research studies) in Cochrane systematic reviews: a review". We do prefer our original title, and note that the manuscript clearly defines what a COS is, but propose this one as an alternative. We defer to the reviewer and the Editor to agree on this point.

Abstract

Page 3, Line 41: This opening sentence needs to be clearer – I have read it a few times and still don't quite understand it. I wonder too whether "core outcome set" needs to be briefly explained (as happens in paragraph two of the Background section) It's not immediately clear what it is.

– The text has been amended.

Methods

Page 3- Lines 45-51. I think this section could be clearer. I don't understand how or why results were compared with existing COS. Its not clear what "CRG outcome selection policies" are. Introducing the notion of a "Summary of Findings" table also adds complexity and confusion. It feels like it is written with an assumption that the reader will understand all of this complexity, but I think it could be written more clearly, in a way that is comprehensible to people unfamiliar with these complexities.

– The text has been amended to make the motivation clearer.

Results:

Page 3- Lines 53- 57. I also find this section very hard to understand – with similar comments to my comments about Methods section of Abstract. For example I have no idea what the sentence means stating "In 7%..." I may well understand all of this clearly by the time I have read the full text – but I think the abstract needs to be much clearer.

– The text has been amended.

Similar concerns about clarity and comprehension for "Strengths and Weaknesses"

– Amended in the manuscript.

Background

Page 5- Line 91. Is it possible to say why this affects the robustness of conclusions? In order to add clarity and underscore significance of this study.

– Further explanation has now been provided.

First paragraph: Page 5, Lines 88-96. There is too much going on in this paragraph that needs more explanation.

– Further explanation has now been provided.

The Background section needs to be clearer and be able to impart clearly the process of a review, how/why reviewers chose certain outcomes etc

– We have added text to the Background to address these comments.

Methods

Page 7 – Lines 141-143. This sentence could be clearer – eg what does 'relevant scope' mean? Also it is not clear from the text why the years 2016, and the time frame 2017 or later – were chosen.

– The scope of a COS has now been defined in the Background section.

Page 7 – Lines 146- 147. It is not clear what “compared results” means.

→ The text has been clarified.

Page 7, Line 152 – for this phrase “outcome selection for the review and for its SoF tables” – do you mean for the reviews conducted within their CRG? Its not clear.

→ The text has been clarified.

Page 7 – Lines 153 -155 and Figure 1. Forgive me but I am uncertain why there are so many Core Outcome Sets within each condition category – and how the many COS interact with each other. I can presume why there are so many – but perhaps it might be explained more clearly in the text. Perhaps I have misunderstood the meaning/purpose of COS – and perhaps in the revision this can be explained more clearly and simply.

→ The scope of a COS has now been defined in the Background section, and an explanation provided as to why this results in there being multiple COS within a broad condition category such as displayed in Figure 1.

Results

Page 10- Lines 208-210. While I have no question about accuracy – as a reader it is a little confusing jumping from the “90” in the first sentence of this paragraph, to the “93” in the next sentence.

→ We have amended the text to clarify this.

Conclusions

Whole tone of the Conclusion needs to change to become more like an academic paper and less like a promotional piece promoting the value of COS.

→ We have revised the text in the Conclusions section.

Reviewer: 3

Reviewer Name: Spyridon N. Papageorgiou

Institution and Country: University of Zurich, Switzerland Please state any competing interests or state 'None declared': None declared

Please leave your comments for the authors below This is a simple and interesting paper on an interesting matter. I have written a few things that are very minor and am supportive of the authors' work. Thanks for letting me see this.

- Short clarification on the sample size used: I take it that 100 was used arbitrarily or were there practical reason. Maybe add a couple of words to clarify this.

→ We have amended the text to clarify this.

- In “(i) mentioned using a COS to choose outcomes for the review”: are we talking about the protocol stage of the review or the write-up (including post hoc adaptations of the registered protocol).

→ This has been clarified in the text.

- Can you elaborate what is meant in (ii) with problems in outcome inconsistency?

→ This has been clarified in the text.

- Regarding the ‘data available on request’ on page 7: I generally am not a great fan of such statements. You might consider if this info can fit in an online appendix and provided a priori.
 - We will defer to the Editor here.

- Maybe you can add a short sentence explaining how is each CRG editor involved in what phases of a review, so that readers not familiar with the structure of Cochrane can understand it better?
 - This has been clarified in the text.

- I am skeptical about the use of 2016 or 2017 as chosen years for the need for a COS to have been published in order for the reviewer to be able to include them. Is it based on an average preparation duration for a review of 2 years (as you searched for reviews published in 2019)? And even if that is true, it might be that the reviewers might not have seen a published COS at the protocol phase, but they could have seen it at a later stage during analysis or write-up. Anyway, just to give this some consideration as an issue in the manuscript, if possible.
 - The COS had to have been published ‘in 2016 or earlier’ rather than 2017. We have acknowledged this as an assumption, and added text to the Discussion section.

- “Of 34 reviews where a COS was available (2016 or earlier), one was used in 7 (21%).”: I found the placement of this statement here somewhat confusing (then I understood that it adds up the 27 reviews from directly above to the 7 reviews at the top of the page...Maybe consider modifying it to make it more straightforward?
 - We have amended the text to clarify this.

- Ok, this is a standard term, but maybe explain the IQR in its first mention.
 - Done.

- There was some vagueness as to where the Results section and the Discussion section ends/starts, but I don’t know exactly the journal’s rules. I personally did not find this a problem and thought the text flow was smooth from reporting of results to discussion of issues. I am also not that sure how much pertinent literature exists in this specific field (I take it not that much and the authors are probably already aware of it).
 - To our knowledge this is the first assessment of Cochrane reviews (and indeed any systematic review) for uptake of COS. There has been limited assessment of COS uptake generally, but we have now included reference to a recent evaluation of COS uptake in a cohort of funded trials in the UK.

- I believe the 1st figure with the COSs is difficult to be seen on paper without magnification option.
 - Figure 1 looks clear to us on the full-page image we submitted. The files seems to have been combined into one submission, in which the figure has been shrunk and fills less than half a page. We defer to the Editor for advice.

- For the next 2 figures, maybe consider differentiating the two years with different columns or colour?
 - We have amended the figures using colour.

VERSION 2 – REVIEW

REVIEWER	Ray Moynihan Bond University Australia
REVIEW RETURNED	15-Jun-2020

GENERAL COMMENTS	Thanks for opportunity to re-review. The paper is much improved. I have recommended acceptance, but I think the clarity could still be improved. I suggest the authors re-read the abstract and manuscript imagining a reader with no knowledge of COS or Cochrane, and try to maximise clarity. One tiny specific comment - I think the lack of a protocol - which is mentioned in the methods - should also be mentioned in the Limitations section.
--

REVIEWER	Spyridon N. Papageorgiou University of Zurich, Switzerland
REVIEW RETURNED	25-May-2020

GENERAL COMMENTS	Nothing more to comment. Some initial comments of mine have been deferred to the editor, so I don't bother with them.
---

VERSION 2 – AUTHOR RESPONSE

In response to the comments from reviewer 2, we have made edits to the abstract and the manuscript to improve the clarity. We have also added the lack of protocol to the limitations section. These edits are highlighted in the marked copy of the manuscript.